# Hepatic Arterial Infusion Chemotherapy with Cisplatin versus Sorafenib for Intrahepatic Advanced Hepatocellular Carcinoma: A Propensity Score-Matched Analysis

**DOI:** 10.3390/cancers13215282

**Published:** 2021-10-21

**Authors:** Yuki Zaizen, Masahito Nakano, Kazuta Fukumori, Yoichi Yano, Kota Takaki, Takashi Niizeki, Kotaro Kuwaki, Masaru Fukahori, Takahiko Sakaue, Sohei Yoshimura, Mika Nakazaki, Ryoko Kuromatsu, Shusuke Okamura, Hideki Iwamoto, Shigeo Shimose, Tomotake Shirono, Yu Noda, Naoki Kamachi, Hironori Koga, Takuji Torimura

**Affiliations:** 1Division of Gastroenterology, Department of Medicine, School of Medicine, Kurume University, Kurume 830-0011, Japan; zaizen_yuki@med.kurume-u.ac.jp (Y.Z.); fukumori_kazuta@kurume-u.ac.jp (K.F.); takaki_kouta@med.kurume-u.ac.jp (K.T.); niizeki_takashi@kurume-u.ac.jp (T.N.); kuwaki_koutarou@kurume-u.ac.jp (K.K.); fukahori_masaru@med.kurume-u.ac.jp (M.F.); sakaue_takahiko@med.kurume-u.ac.jp (T.S.); yoshimura_souhei@med.kurume-u.ac.jp (S.Y.); nomiyama_mika@med.kurume-u.ac.jp (M.N.); ryoko@med.kurume-u.ac.jp (R.K.); okamura_shyuusuke@kurume-u.ac.jp (S.O.); iwamoto_hideki@med.kurume-u.ac.jp (H.I.); shimose_shigeo@med.kurume-u.ac.jp (S.S.); shirono_tomotake@med.kurume-u.ac.jp (T.S.); noda_yuu@med.kurume-u.ac.jp (Y.N.); kamachi_naoki@med.kurume-u.ac.jp (N.K.); hirokoga@med.kurume-u.ac.jp (H.K.); tori@med.kurume-u.ac.jp (T.T.); 2Division of Gastroenterology, Department of Medicine, Japan Community Health Care Organization, Saga Central Hospital, Saga 849-8522, Japan; syano@kumin.ne.jp; 3Division of Gastroenterology, Department of Medicine, Ōmuta City Hospital, Ōmuta 836-8567, Japan

**Keywords:** hepatocellular carcinoma, hepatic arterial infusion chemotherapy, cisplatin, sorafenib, multikinase inhibitor, risk factors, propensity score-matched analysis

## Abstract

**Simple Summary:**

Thus far, clinical studies have shown that immunotherapy (atezolizumab-bevacizumab) has shown better and favorable overall survival than sorafenib for advanced hepatocellular carcinoma (HCC). However, the treatment outcomes of hepatic arterial infusion chemotherapy (HAIC) with cisplatin compared with sorafenib for intrahepatic advanced HCC remain unclear. We therefore aimed to determine the prognostic factors for HAIC with cisplatin. Our results showed that HAIC with cisplatin could significantly prolong the overall survival for intrahepatic advanced HCC and had a longer prognostic effect than sorafenib. Therefore, our results suggest that HAIC should be used in intrahepatic advanced HCC.

**Abstract:**

Given that the outcome of hepatic arterial infusion chemotherapy (HAIC) with cisplatin for intrahepatic advanced hepatocellular carcinoma (HCC) is unclear, we aimed to compare prognostic factors for overall survival (OS) following HAIC with cisplatin versus sorafenib for intrahepatic advanced HCC using propensity score-matched analysis. We enrolled 331 patients with intrahepatic advanced HCC who received HAIC with cisplatin (n = 88) or sorafenib (n = 243) between June 2006 and March 2020. No significant difference was observed in OS between HAIC with cisplatin and sorafenib cohorts (median survival time [MST]: 14.0 vs. 12.3 months; *p* = 0.0721). To reduce confounding effects, 166 patients were selected using propensity score-matched analysis (n = 83 for each treatment). HAIC with cisplatin significantly prolonged OS compared with sorafenib (MST: 15.6 vs. 11.0 months; *p* = 0.0157). Following stratification according to the Child-Pugh classification, for patients with class A (MST: 24.0 vs. 15.0 months; *p* = 0.0145), HAIC with cisplatin rather than sorafenib significantly prolonged OS. Our findings suggest that HAIC with cisplatin demonstrates longer prognostic effects than sorafenib in intrahepatic advanced HCC.

## 1. Introduction

Liver cancer was the sixth most commonly diagnosed cancer and the fourth leading cause of cancer-related deaths worldwide in 2018, with an estimated 841,000 new cases and 782,000 deaths [1,2,3,4]. Liver cancer includes hepatocellular carcinoma (HCC) that accounts for 75–85% of all liver cancer cases [1,2]. Early-stage HCC may be curable radically via hepatic resection, radiofrequency ablation, or liver transplantation; however, patients with advanced HCC have a poor prognosis [5,6].

Hepatic arterial infusion chemotherapy (HAIC) is a treatment option for advanced HCC [7]. Theoretically, HAIC can increase the concentrations of the anticancer drug in the liver and consequently reduce the occurrence of systemic adverse events caused by the anticancer drug [8]. Recently, there has been accumulating evidence regarding the efficacy of HAIC for treating advanced HCC [9,10,11].

The use of molecular-targeted agents (MTAs) is another treatment option for advanced HCC [7]. MTAs such as sorafenib were approved as first-line treatment for advanced HCC based on the results of two studies, namely the Sorafenib Hepatocellular carcinoma Assessment Randomized Protocol (SHARP) study [12] and the Asia-Pacific study [13]; these studies reported superior survival outcomes with sorafenib over those with placebo. In addition, immunotherapy, which is based on the combination of atezolizumab and bevacizumab, resulted in better outcomes than sorafenib when used as the first-line treatment for advanced HCC [14].

In a randomized phase II trial, treatment with sorafenib plus HAIC with cisplatin yielded favorable overall survival (OS) compared with treatment with sorafenib alone in patients with advanced HCC [15]. However, treatment outcomes of HAIC with cisplatin versus those of sorafenib for advanced HCC remain unclear. Therefore, in this study, we aimed to determine the prognostic effects of HAIC with cisplatin and the associated OS duration compared with those of sorafenib for advanced HCC. In view of this, to reduce confounding effects, we performed propensity score-matched analysis.

## 2. Materials and Methods

### 2.1. Ethical Approval

The study was approved by the Ethics Committee of Kurume University (No. 10009) and Saga Central Hospital (No. 21002) and was conducted according to the guidelines of the 1975 Declaration of Helsinki.

### 2.2. Diagnosis

HCC was either confirmed histologically or diagnosed using noninvasive criteria according to the European Association for the Study of Liver [16]. Intrahepatic lesions and vascular invasion were diagnosed using a combination of imaging techniques such as contrast-enhanced computed tomography, magnetic resonance imaging, ultrasonography, and digital subtraction angiography. Additionally, serum levels of alpha-fetoprotein (AFP) and des-gamma-carboxy prothrombin (DCP) were measured for up to 1 month before treatment. The presence of intra-abdominal metastases was detected on abdominal computed tomography, magnetic resonance imaging, and ultrasonography, which were performed to evaluate intrahepatic lesions. Liver function was evaluated using both the Child-Pugh classification and albumin-bilirubin (ALBI) score [17]. Tumor stage was determined according to the Barcelona Clinic Liver Cancer (BCLC) staging classification [18,19].

### 2.3. Patients Receiving HAIC with Cisplatin

Since the approval of cisplatin (DDP-H, IA-Call, Nippon Kayaku, Tokyo, Japan) use for advanced HCC in Japan, we treated 98 patients for advanced HCC with HAIC and cisplatin in Saga Central Hospital between July 2006 and March 2020. One patient with extrahepatic metastasis and nine patients with BCLC stage A were excluded; therefore, we enrolled 88 consecutive patients who were diagnosed with intrahepatic advanced HCC who received HAIC with cisplatin.

After conventional visceral angiography, HAIC was administered by introducing an angiographic catheter into the proper, right, or left hepatic artery or the branched feeding artery using Seldinger’s technique and not using any implanted port system for HAIC. Cisplatin was dissolved in saline solution and heated to 50 °C, and was then injected at a dose of 65 mg/m^2^ over 20–40 min without lipiodol and gelatin sponge. Until the appearance of tumor progression and/or unacceptable toxicity, the treatment was repeated every 2–3 months for a maximum of 26 cycles. All patients had antiemetic prophylaxis with a 5-HT_3_ antagonist (granisetron 1 mg) and received adequate hydration and diuretics for protection against cisplatin-induced renal dysfunction.

### 2.4. Patients Receiving Sorafenib

Eligibility criteria for this study were similar to those for the SHARP study [12], as follows: (1) Eastern Cooperative Oncology Group performance status of 0–1, (2) measurable disease using the Response Evaluation Criteria in Solid Tumors, (3) Child-Pugh class A or B, (4) leukocyte count ≥ 2000/mm^3^, (5) platelet count ≥ 50 × 10^9^/L, (6) hemoglobin level ≥ 8.5 g/dL, (7) serum creatinine level < 1.5 mg/dL, and (8) no ascites or encephalopathy. Since the approval of sorafenib use for advanced HCC in Japan, we treated 553 patients for advanced HCC with sorafenib in 19 participating institutions of the Kurume Liver Cancer Study Group of Japan between May 2010 and March 2020. Among these patients, 302 patients with extrahepatic metastasis and eight patients with BCLC stage A were excluded; therefore, we enrolled 243 consecutive patients who were diagnosed with intrahepatic advanced HCC and received sorafenib.

### 2.5. Treatment Outcome

The treatment outcome of this study was OS, which was defined as the time from the initiation of HAIC with cisplatin or sorafenib to the date of death or the patient’s last follow-up.

### 2.6. Statistical Analysis

Baseline patient characteristics were analyzed using descriptive statistical methods: age, albumin level, total bilirubin level, ALBI score, prothrombin time, AFP level, and DCP were calculated using the t-test, and sex, etiology, Child-Pugh class, macrovascular invasion, and BCLC stage were calculated using the chi-square test. Univariate and multivariate Cox proportional hazards analyses were performed to evaluate the interaction between patient characteristics and OS. Survival curves were constructed using the Kaplan-Meier analysis with the log-rank test. Results are expressed as mean ± standard deviation (SD) and median (range) or n (%). A *p*-value of <0.05 was considered to indicate statistical significance. JMP software (version 15; SAS Institute, Inc., Cary, NC, USA) was used for all statistical analyses.

The following 12 variables related to the prognosis of advanced HCC were considered at the start of the follow-up: age, sex, etiology, Child-Pugh class, macrovascular invasion, BCLC stage, albumin level, total bilirubin level, ALBI score, prothrombin time, AFP, and DCP. We used these propensity scores to conduct 1:1 nearest neighbor matching within a caliper of 0.20, as previous studies have shown this SD percentage of the logit of the propensity score to be generally suitable as a caliper for propensity score-matched analysis [20].

## 3. Results

### 3.1. Patient Characteristics

Table 1 shows the characteristics of 331 consecutive patients who were diagnosed with intrahepatic advanced HCC and received either HAIC with cisplatin (n = 88) or sorafenib (n = 243). A higher proportion of patients tested positive for the hepatitis C virus (*p* = 0.0196) and had Child-Pugh class B (*p* = 0.0013) in the HAIC with cisplatin cohort, whereas a higher proportion of patients had macrovascular invasion (*p* = 0.0233) in the sorafenib cohort. The ALBI score (*p* = 0.0006) was higher in the HAIC with cisplatin cohort, whereas albumin levels (*p* = 0.0009) and prothrombin time (*p* < 0.0001) were higher in the sorafenib cohort. Age; sex; BCLC stage; and total bilirubin, AFP, and DCP levels were equivalent between the HAIC with cisplatin and sorafenib cohorts.

Regarding the tumor number, 5 (6%) had single tumors and 83 (94%) had multiple tumors in the HAIC cohort, whereas 15 (6%) had single tumors and 228 (94%) had multiple tumors in the sorafenib cohort. Regarding the tumor size, it was 37.2 ± 30.9 mm in the HAIC cohort and 42.4 ± 23.2 mm in the sorafenib cohort.

### 3.2. Univariate and Multivariate Analyses of OS

Table 2 shows the results of univariate and multivariate analyses of OS. In the HAIC with cisplatin cohort (Table 2A), univariate analyses of OS revealed five variables as prognostic factors: Child-Pugh class (B, *p* = 0.0002), macrovascular invasion (yes, *p* = 0.0002), BCLC stage (C, *p* < 0.0001), ALBI score (median level of ≥−2.11, *p* = 0.0349), and AFP (median level of ≥108 ng/mL, *p* = 0.0472). Multivariate analyses of OS identified a variable as independent prognostic factor: Child-Pugh class (B, *p* = 0.0059). On the other hand, in the sorafenib cohort (Table 2B), univariate analyses of OS revealed six variables as prognostic factors: Child-Pugh class (B, *p* < 0.0001), macrovascular invasion (yes, *p* = 0.0401), BCLC stage (C, *p* = 0.0401), ALBI score (median level of ≥−2.31, *p* < 0.0001), AFP (median level of ≥93 ng/mL, *p* < 0.0001), and DCP (median level of ≥549 mAU/mL, *p* < 0.0001). Multivariate analyses of OS identified three variables as independent prognostic factors: Child-Pugh class (B, *p* = 0.0058), ALBI score (median level of ≥−2.31, *p* = 0.0039), and AFP (median level of ≥93 ng/mL, *p* < 0.0001).

### 3.3. Survival Outcomes

Figure 1 shows the results of the Kaplan-Meier analysis of OS with the log-rank test between the HAIC with cisplatin and sorafenib cohorts. The median survival time (MST) was 14.0 months in the HAIC with cisplatin cohort (blue line, n = 88) and 12.3 months in the sorafenib cohort (red line, n = 243) (*p* = 0.0721). OS did not differ significantly between both cohorts.

In the HAIC group, there were 68 deaths (77%) and 20 survivals (23%). Meanwhile, in the sorafenib group, there were 206 deaths (85%) and 27 survivals (11%), and the details of 10 cases (4%) were unknown.

### 3.4. Propensity Score-Matched Analysis

To reduce confounding effects, we performed propensity score-matched analysis to match patients treated with HAIC with cisplatin (n = 88) with those treated with sorafenib (n = 243) [21,22]. The propensity scores (mean ± SD) of the patients treated with HAIC with cisplatin and sorafenib were 0.834 ± 5.694 and −4.598 ± 8.444, respectively. Based on the results of propensity score-matched analysis, 166 patients were selected (HAIC with cisplatin, n = 83; sorafenib, n = 83). Following the propensity score-matched analysis, the propensity scores (mean ± SD) of the patients treated with HAIC with cisplatin and sorafenib were −0.431 ± 0.863 and −0.538 ± 0.779, respectively.

### 3.5. Characteristics of Patients Diagnosed with HCC Following Propensity Score-Matched Analysis

Table 3 shows the characteristics of 166 patients who were diagnosed with intrahepatic advanced HCC and received HAIC with cisplatin (n = 83) or sorafenib (n = 83) following propensity score-matched analysis. No significant differences were observed in any variables between the HAIC with cisplatin and sorafenib cohorts using propensity score-matched analysis.

### 3.6. Transition of Treatment Following Propensity Score-Matched Analysis

Thirteen patients received PRFA, 2 received PEIT, 16 received HAIC with the reservoir system, 25 received TACE, and 7 received hepatic resection as the previous treatment, in the sorafenib cohort. Information on previous therapy before HAIC could not be evaluated. There was no crossover between the HAIC and sorafenib cohorts. Secondary treatment was not required in the HAIC cohort. Meanwhile, in the sorafenib cohort, 11 patients received other MTAs and 32 patients received HAIC with the reservoir system or TACE as the secondary treatment. The total dose (mg) in the HAIC cohort was 308.9 ± 318.3 and 69929.1 ± 97478.5 in the sorafenib cohort.

### 3.7. Univariate and Multivariate Analyses of OS Following Propensity Score-Matched Analysis

Table 4 shows the results of univariate and multivariate analyses of OS following propensity score-matched analysis. Univariate analyses of OS revealed seven variables as prognostic factors: Child-Pugh class (*p* < 0.0001), macrovascular invasion (*p* = 0.0003), BCLC stage (*p* = 0.0003), ALBI score (*p* < 0.0001), AFP (*p* = 0.0002), DCP (*p* < 0.0001), and treatment (*p* = 0.0164). Multivariate analyses of OS identified six variables as independent prognostic factors: Child-Pugh class (*p* = 0.0119), macrovascular invasion (*p* = 0.0180), BCLC stage (*p* = 0.0180), AFP (*p* = 0.0164), DCP (*p* = 0.0162), and treatment (*p* = 0.0090). Treatment with HAIC was a significant prognostic factor in both univariate and multivariate analyses of OS following propensity score-matched analysis.

### 3.8. Survival Outcomes Following Propensity Score-Matched Analysis

Figure 2 shows the results of the Kaplan-Meier analysis of OS with the log-rank test between the HAIC with cisplatin and sorafenib cohorts following propensity score-matched analysis. The MST was 15.6 months in the HAIC with cisplatin cohort (blue line, n = 83) and 11.0 months in the sorafenib cohort (red line, n = 83) (*p* = 0.0157). The HAIC with cisplatin cohort demonstrated significantly better outcomes with regard to OS than the sorafenib cohort.

### 3.9. Survival Outcomes Having Child-Pugh Class A Following Propensity Score-Matched Analysis

Figure 3 shows the results of the Kaplan-Meier analysis of OS with the log-rank test between the HAIC with cisplatin and sorafenib cohorts having Child-Pugh class A, following propensity score-matched analysis. For patients with Child-Pugh class A, the MST was 24.0 months in the HAIC with cisplatin cohort (blue line, n = 53) and 15.0 months in the sorafenib cohort (red line, n = 59) (*p* = 0.0145). The HAIC with cisplatin exhibited significantly better outcomes in the Child-Pugh class A cohort with regard to OS than in the sorafenib cohort.

### 3.10. Changes in Hepatic Reserve Factor before and 1 Month after Treatment Following Propensity Score-Matched Analysis

We compared the ALBI scores before treatment and 1 month after treatment as a simple index of hepatic reserve factor. In the HAIC cohort, it was −2.09 ± 0.48 before treatment and −2.08 ± 0.57 at 1 month after treatment (*p* = 0.9429). On the other hand, in the sorafenib cohort, it was −2.14 ± 0.44 before treatment and −2.03 ± 0.52 at 1 month after treatment (*p* = 0.1792). There was no significant difference in the changes in hepatic reserve factor before and after treatment between the two cohorts.

### 3.11. Adverse Events Following Propensity Score-Matched Analysis

There were few noticeable adverse events leading to discontinuation of treatment in the HAIC group—one patient had anaphylactic shock as an adverse event that led to discontinuation of treatment. Meanwhile, in the sorafenib cohort, 73 patients (88%) had adverse events, comprising 28 patients with hand-foot skin reaction, 17 with liver dysfunction, 15 with diarrhea, 10 with loss of appetite, and 5 with hypertension.

## 4. Discussion

In this study, we assessed the OS of patients with intrahepatic advanced HCC between the HAIC with cisplatin and sorafenib cohorts. The results showed that the OS did not differ significantly between both cohorts for intrahepatic advanced HCC, among the enrolled patients (Figure 1). However, in the HAIC with cisplatin cohort, tumor factors were significantly better, whereas in the sorafenib cohort, the hepatic reserve factor was significantly better (Table 1). To reduce confounding effects, we performed propensity score-matched analysis to match patients treated with HAIC with cisplatin and those treated with sorafenib (Table 3). HAIC with cisplatin resulted in significantly better outcomes with regard to OS than sorafenib following propensity score-matched analysis (Figure 2). Our results suggest that HAIC, rather than sorafenib, should be used for intrahepatic advanced HCC without extrahepatic metastasis. There are several treatment strategies for advanced HCC, such as transarterial chemoembolization (TACE), HAIC, and systemic therapy. There has been a consensus on the management of advanced HCC with extrahepatic metastasis—systemic therapy (MTAs or immunotherapy, such as the combination of atezolizumab and bevacizumab) should be used [14]. In contrast, for managing intrahepatic advanced HCC without extrahepatic metastasis, the optimal choice remains controversial.

Fundamentally, the indication for administering sorafenib in the management of advanced HCC is only Child-Pugh class A [12]. Therefore, we stratified patients according to their Child-Pugh class following propensity score-matched analysis. Following stratification according to Child-Pugh class, for patients with Child-Pugh class A, HAIC with cisplatin showed significantly better outcomes with regard to OS than sorafenib (Figure 3). Our results suggest that HAIC should be used for treating intrahepatic advanced HCC without extrahepatic metastasis. In particular, for patients with Child-Pugh class A, both HAIC and sorafenib are indicated for this condition; therefore, our results suggest that HAIC, not sorafenib, should be used in intrahepatic advanced HCC with Child-Pugh class A.

Moreover, using univariate and multivariate analyses, we assessed prognostic factors for intrahepatic advanced HCC managed in all enrolled patients (Table 2). Multivariate analyses of OS revealed a variable as independent prognostic factor: Child-Pugh class (B) in the HAIC with cisplatin cohort (Table 2A), on the other hand, three variables as independent prognostic factors: Child-Pugh class (B), higher serum ALBI levels, and higher serum AFP levels in the sorafenib cohort (Table 2B). It is well-known that the hepatic reserve factor and tumor factor contribute to OS of patients with HCC, which is consistent with the finding observed in the present study [23,24,25].

For managing intrahepatic advanced HCC, TACE or HAIC has been widely used for obtaining a higher antitumor effect as they evenly distribute the anticancer drug through the hepatic artery [26]. However, TACE involves inserting a microcatheter selectively into the tumor-feeding artery; this requires high-level skills and adequate treatment time. In HAIC with the reservoir system, to place an implantable port system, it is necessary to place a catheter in the appropriate position and embolize the blood vessels with a coil so that the anticancer drug is delivered only to HCC; this also requires high-level skills and adequate treatment time [27]. In this study, we administered HAIC with cisplatin only by introducing the angiographic catheter into the proper, right, or left hepatic artery or the branched feeding artery using Seldinger’s technique and then injecting the anticancer drug. The HAIC with cisplatin method is more convenient than TACE or HAIC with the reservoir system. In particular, HAIC with cisplatin shortens the treatment time, which reduces the radiation exposure and physical burden on the patients compared with TACE or HAIC with the reservoir system. In this study, many patients were able to continue multiple cycles of HAIC (mean, 5.2 ± 4.8 cycles) for up to 26 cycles. One reason for this was the occurrence of few adverse events, which made it possible to continue HAIC. Therefore, our results suggest that HAIC has advantages of being simple to use, shortening the treatment time, and resulting in only few adverse events.

Two randomized controlled trials (RCTs) have verified the additional effect of HAIC over sorafenib for managing advanced HCC [28,29]. One study described that the addition of HAIC to sorafenib did not significantly improve the OS of patients with advanced HCC [28], whereas another study described that sorafenib plus HAIC improved OS compared to sorafenib alone in patients with HCC and portal vein invasion [29]. However, several non-RCTs have revealed that HAIC improved OS compared to sorafenib in patients with advanced HCC [30,31,32]. Therefore, RCTs comparing HAIC and sorafenib in patients with advanced HCC should be conducted.

Our current study has some limitations. First, regarding the HAIC with cisplatin cohort, our study had a single-center retrospective design with a relatively small sample size (n = 88) for intrahepatic advanced HCC. Second, the treatment (HAIC with cisplatin or sorafenib) was selected at the discretion of the chief physician, and patients were not randomized after receiving approval for sorafenib use. This resulted in a selection bias for patients with advanced HCC. Lastly, the therapeutic effects in all cases and the information of previous therapy before HAIC could not be evaluated. Therefore, a multicenter prospective study with a larger patient population should be conducted in the future.

## 5. Conclusions

HAIC demonstrated significantly better outcomes with regard to OS than sorafenib following propensity score-matched analysis. Our results suggest that HAIC should be used rather than sorafenib in intrahepatic advanced HCC cases without extrahepatic metastasis.

## Figures and Tables

**Figure 1 cancers-13-05282-f001:**
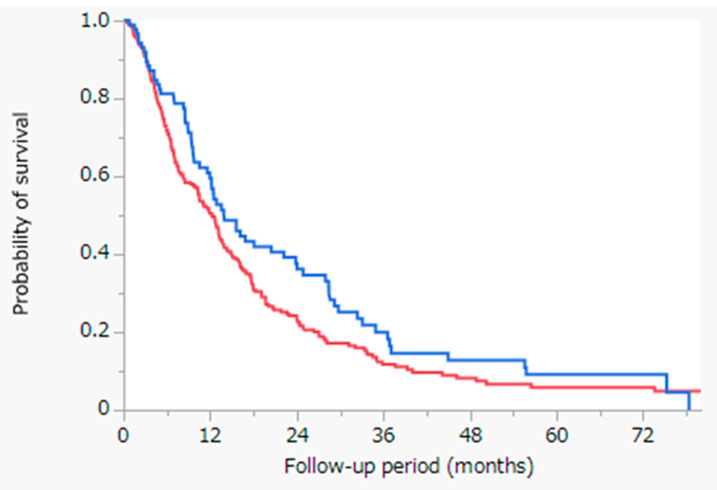
Kaplan-Meier analysis of OS with the log-rank test between the HAIC with cisplatin and sorafenib cohorts. Blue line: HAIC with cisplatin cohort (n = 88), MST = 14.0 months; red line: sorafenib cohort (n = 243), MST = 12.3 months; *p* = 0.0721. Abbreviations: OS = overall survival; HAIC = hepatic arterial infusion chemotherapy; MST = median survival time.

**Figure 2 cancers-13-05282-f002:**
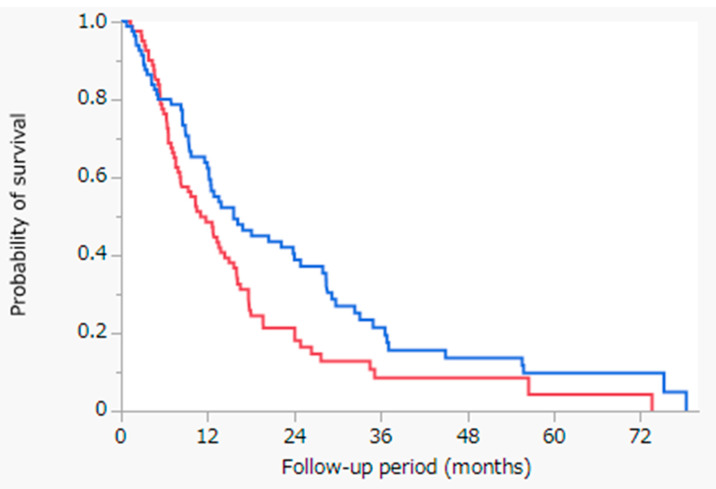
Kaplan-Meier analysis of OS with the log-rank test between the HAIC with cisplatin and sorafenib cohorts following propensity score-matched analysis. Blue line: HAIC with cisplatin cohort (n = 83), MST = 15.6 months; Red line: sorafenib cohort (n = 83), MST = 11.0 months; *p* = 0.0157. Abbreviations: OS = overall survival; HAIC = hepatic arterial infusion chemotherapy; MST = median survival time.

**Figure 3 cancers-13-05282-f003:**
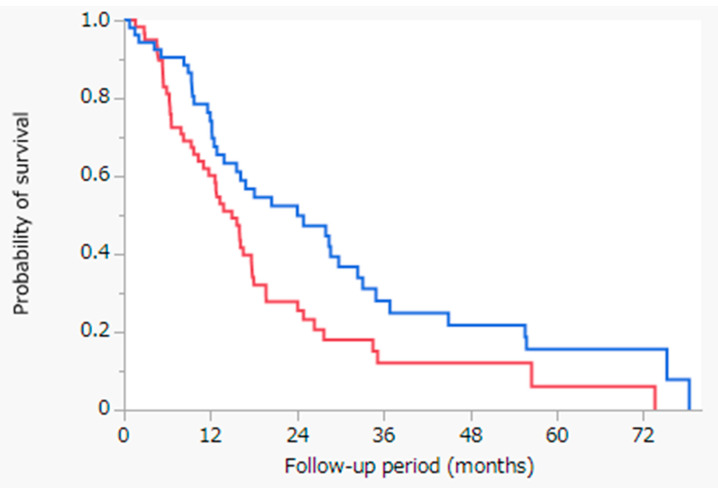
Kaplan-Meier analysis of the OS with the log-rank test between the HAIC with cisplatin and sorafenib cohorts having Child-Pugh class A, following propensity score-matched analysis. Blue line: HAIC with cisplatin cohort having Child-Pugh class A (n = 53), MST = 24.0 months; red line: sorafenib cohort having Child-Pugh class A (n = 59), MST = 15.0 months; *p* = 0.0145. Abbreviations: OS = overall survival; HAIC = hepatic arterial infusion chemotherapy; MST = median survival time.

**Table 1 cancers-13-05282-t001:** Patient characteristics (n = 331).

Variable	HAIC (n = 88)	Sorafenib (n = 243)	*p*-Value
Age (years)	73.8 ± 9.6	72.4 ± 9.5	0.2515
	75.2 (47.8–88.6)	72.8 (35.7–94.4)	
Sex (male/female)	61 (69%)/27 (31%)	193 (79%)/50 (21%)	0.0545
Etiology (HBV/HCV/HBV + HCV/both negative)	6 (7%)/71 (81%)/0 (0%)/11 (12%)	37 (15%)/153 (63%)/3 (1%)/50 (21%)	0.0196
Child-Pugh class (A/B)	56 (64%)/32 (36%)	196 (81%)/47 (19%)	0.0013
Macrovascular invasion (yes/no)	13 (15%)/75 (85%)	65 (27%)/178 (73%)	0.0233
BCLC stage (B/C)	71 (81%)/17 (19%)	178 (73%)/65 (27%)	0.1665
Albumin (g/dL)	3.4 ± 0.5	3.6 ± 0.5	0.0009
	3.5 (2.2–4.4)	3.6 (2.1–4.8)	
Total bilirubin level (mg/dL)	1.0 ± 0.5	0.9 ± 0.5	0.2451
	0.9 (0.3–2.7)	0.9 (0.2–3.4)	
ALBI score	−2.08 ± 0.48	−2.28 ± 0.47	0.0006
	−2.11 (−2.99–−0.83)	−2.31 (−3.28–−1.02)	
Prothrombin time (%)	75.3 ± 11.0	85.0 ± 16.4	<0.0001
	74.5 (44.6–105.5)	84.4 (23.0–130.0)	
AFP (ng/mL)	4399 ± 24,315	7275 ± 49,351	0.6009
	108 (2–222,500)	93 (1–720,500)	
DCP (mAU/mL)	8914 ± 38,872	9824 ± 31,329	0.8276
	290 (6–344,000)	548 (8–335,810)	

Abbreviations: HAIC = hepatic arterial infusion chemotherapy, HBV = hepatitis B virus, HCV = hepatitis C virus, BCLC = Barcelona Clinic Liver Cancer, ALBI = albumin-bilirubin, AFP = alpha-fetoprotein, DCP = Des-gamma-carboxy prothrombin. Results are expressed as mean ± standard deviation and median (range) or n (%).

**Table 2 cancers-13-05282-t002:** (**A**). Univariate and multivariate analyses of OS in HAIC with cisplatin cohort (n = 88). (**B**). Univariate and multivariate analyses of OS in the sorafenib cohort (n = 243).

(**A**)				
**Variable**	**Univariate Analysis**		**Multivariate Analysis**	
	**HR (95% CI)**	***p*-Value**	**HR (95% CI)**	***p*-Value**
Age (≥75.3 years)	0.799 (0.492–1.297)	0.3633	0.588 (0.331–1.043)	0.0669
Sex (Male)	0.926 (0.551–1.557)	0.7727	1.023 (0.576–1.814)	0.9388
Etiology (HCV)	1.116 (0.596–2.090)	0.7279	1.407 (0.709–2.795)	0.3147
Child-Pugh class (B)	2.752 (1.657–4.571)	0.0002	2.848 (1.318–6.152)	0.0059
Macrovascular invasion (Yes)	4.305 (2.196–8.437)	0.0002	1.747 (0.485–6.297)	0.3810
BCLC stage (C)	4.027 (2.152–7.536)	<0.0001	2.680 (0.847–8.478)	0.1240
ALBI score (≥−2.11)	1.683 (1.037–2.730)	0.0349	0.781 (0.380–1.605)	0.4960
AFP (≥108 ng/mL)	1.646 (1.008–2.688)	0.0472	1.776 (0.973–3.243)	0.0598
DCP (≥290 mAU/mL)	1.593 (0.971–2.615)	0.0680	1.682 (0.978–2.894)	0.0619
(**B**)				
**Variable**	**Univariate Analysis**		**Multivariate Analysis**	
	**HR (95% CI)**	***p*-Value**	**HR (95% CI)**	***p*-Value**
Age (≥72.8 years)	1.141 (0.868–1.501)	0.3438	1.269 (0.938–1.718)	0.1224
Sex (Male)	0.714 (0.506–1.008)	0.0639	0.884 (0.599–1.305)	0.5390
Etiology (HCV)	1.169 (0.878–1.557)	0.2809	0.974 (0.711–1.335)	0.8715
Child-Pugh class (B)	2.541 (1.805–3.578)	<0.0001	1.794 (1.195–2.692)	0.0058
Macrovascular invasion (Yes)	1.388 (1.022–1.886)	0.0401	1.085 (0.772–1.526)	0.6397
BCLC stage (C)	1.388 (1.022–1.886)	0.0401	1.085 (0.772–1.526)	0.6397
ALBI score (≥−2.31)	2.077 (1.567–2.753)	<0.0001	1.655 (1.179–2.323)	0.0039
AFP (≥93 ng/mL)	2.289 (1.725–3.036)	<0.0001	2.091 (1.512–2.893)	< 0.0001
DCP (≥548 mAU/mL)	1.912 (1.446–2.529)	<0.0001	1.260 (0.920–1.726)	0.1483

Abbreviations: OS = overall survival, HAIC = hepatic arterial infusion chemotherapy, HR = hazard ratio, CI = confidence interval, HCV = hepatitis C virus, BCLC = Barcelona Clinic Liver Cancer, ALBI = albumin-bilirubin, AFP = alpha-fetoprotein, DCP = des-gamma-carboxy prothrombin.

**Table 3 cancers-13-05282-t003:** Patient characteristics following propensity score-matched analysis (n = 166).

Variable	HAIC (n = 83)	Sorafenib (n = 83)	*p*-Value
Age (years)	73.7± 9.6	73.2 ± 10.0	0.7394
	74.8 (47.8–88.6)	74.3 (35.7–91.6)	
Sex (male/female)	56 (67%)/27 (33%)	60 (72%)/23 (28%)	0.4986
Etiology (HBV/HCV/HBV + HCV/both negative)	6 (7%)/67 (81%)/0 (0%)/10 (12%)	7 (9%)/65 (78%)/0 (0%)/11 (13%)	0.9225
Child-Pugh class (A/B)	53 (64%)/30 (36%)	59 (71%)/24 (29%)	0.3202
Macrovascular invasion (yes/no)	12 (14%)/71 (86%)	9 (11%)/74 (89%)	0.4836
BCLC stage (B/C)	71 (86%)/12 (14%)	74 (89%)/9 (11%)	0.4836
Albumin level (g/dL)	3.4 ± 0.5	3.5 ± 0.5	0.4229
	3.5 (2.2–4.4)	3.5 (2.6–4.6)	
Total bilirubin (mg/dL)	1.0 ± 0.5	1.0 ± 0.4	0.7646
	0.9 (0.3–2.7)	1.0 (0.3–2.9)	
ALBI score	−2.09 ± 0.48	−2.14 ± 0.44	0.4712
	−2.11 (−2.99–−0.83)	−2.16 (−3.23–−1.26)	
Prothrombin time (%)	75.0 ± 10.4	76.1 ± 12.0	0.5602
	74.5 (44.6–100.7)	78.0 (39.0–97.0)	
AFP (ng/mL)	4462 ± 25,019	4396 ± 19,661	0.9849
	75 (2–222,500)	100 (3–177,630)	
DCP (mAU/mL)	8562 ± 39,732	7898 ± 20,603	0.8928
	316 (6–344,000)	582 (11–112,000)	

Abbreviations: HAIC = hepatic arterial infusion chemotherapy, HBV = hepatitis B virus, HCV = hepatitis C virus, BCLC = Barcelona Clinic Liver Cancer, ALBI = albumin-bilirubin, AFP = alpha-fetoprotein, DCP = Des-gamma-carboxy prothrombin. Results are expressed as mean ± standard deviation and median (range) or n (%).

**Table 4 cancers-13-05282-t004:** Results of univariate and multivariate analyses of OS following propensity score-matched analysis (n = 166).

Variable	Univariate Analysis		Multivariate Analysis	
	HR (95% CI)	*p*-Value	HR (95% CI)	*p*-Value
Age (≥74.4 years)	0.887 (0.629–1.251)	0.4942	0.855 (0.589–1.242)	0.4103
Sex (male)	0.785 (0.541–1.139)	0.2096	0.922 (0.612–1.388)	0.6976
Etiology (HCV)	1.234 (0.811–1.952)	0.3354	1.397 (0.871–2.328)	0.1702
Child-Pugh class (B)	2.615 (1.799–3.761)	<0.0001	1.783 (1.136–2.807)	0.0119
Macrovascular invasion (Yes)	2.813 (1.699–4.655)	0.0003	2.035 (1.136–3.557)	0.0180
BCLC stage (C)	2.813 (1.699–4.655)	0.0003	2.035 (1.164–3.557)	0.0180
ALBI score (≥−2.14)	2.121 (1.500–3.009)	<0.0001	1.501 (0.968–2.308)	0.0692
AFP (≥97 ng/mL)	1.954 (1.378–2.777)	0.0002	1.587 (1.089–2.316)	0.0164
DCP (≥491 mAU/mL)	2.061 (1.448–2.939)	<0.0001	1.597 (1.091–2.339)	0.0162
Treatment (HAIC)	0.653 (0.460–0.925)	0.0164	0.604 (0.413–0.883)	0.0090

Abbreviations: OS = overall survival, HAIC = hepatic arterial infusion chemotherapy, HR = hazard ratio, CI = confidence interval, HCV = hepatitis C virus, BCLC = Barcelona Clinic Liver Cancer, ALBI = albumin-bilirubin, AFP = alpha-fetoprotein, DCP = des-gamma-carboxy prothrombin.

## Data Availability

The data that support the findings of this study are available from the corresponding author, M.Nakano, on reasonable request.

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
