# Peer review of "Hepatic Arterial Infusion Chemotherapy with Cisplatin versus Sorafenib for Intrahepatic Advanced Hepatocellular Carcinoma: A Propensity Score-Matched Analysis"

_cancers, 2021, doi:10.3390/cancers13215282_

Round 1

Reviewer 1 Report

I congratulate the authors for the manuscript. Nevertheless, I have some questions: 
- I am irritated by the presentation of the OS startified by Child Score. As the authors themselves state, there is no indication for sorafenib for patients with a score of more than A.  In this respect, in my opinion, the comparison in this regard is superfluous. 

  • The authors defined survival as either certain death or completion of participation in the follow-up.  In view of the differences in OS of a few months, this seems to be a potentially significant source of error.The proportion of patients who died and those who did not participate in the follow-up should be indicated.
  • The technical method of application (only up to the 1st order of the hepatic vessels) seems questionable. The conclusion that this is significantly easier than TACE or more selective procedures is wrong. The conditions for carrying out both procedures are almost the same as far as the consumables are concerned, only the time factor is different. 

Considering the retrospective study design with method-related limitations, the other results are adequately presented. In summary, and especially in view of the existing data from randomised trials, this paper does not provide any new robust evidence on the topic. 

Author Response

I congratulate the authors for the manuscript. Nevertheless, I have some questions: 
- I am irritated by the presentation of the OS startified by Child Score. As the authors themselves state, there is no indication for sorafenib for patients with a score of more than A.  In this respect, in my opinion, the comparison in this regard is superfluous.

Response: We thank the reviewer for this comment. We agree with your opinion on no indication for sorafenib for patients above Child-Pugh class A; hence, we have modified Figure 3 to focus on Child-Pugh class A.

  • The authors defined survival as either certain death or completion of participation in the follow-up.  In view of the differences in OS of a few months, this seems to be a potentially significant source of error. The proportion of patients who died and those who did not participate in the follow-up should be indicated.

Response: We appreciate this recommendation. In the HAIC group, there were 68 deaths (77%) and 20 survivals (23%). On the other hand, in the sorafenib group, there were 206 deaths (85%) and 27 survivals (11%), and the details of 10 cases (4%) were unknown.

  • The technical method of application (only up to the 1st order of the hepatic vessels) seems questionable. The conclusion that this is significantly easier than TACE or more selective procedures is wrong. The conditions for carrying out both procedures are almost the same as far as the consumables are concerned, only the time factor is different. 

Response: We appreciate this comment and agree with the reviewer. HAIC with cisplatin shortens the treatment time, which reduces radiation exposure and physical burden on the patients, compared with TACE or HAIC with the reservoir system.

Considering the retrospective study design with method-related limitations, the other results are adequately presented. In summary, and especially in view of the existing data from randomised trials, this paper does not provide any new robust evidence on the topic. 

Reviewer 2 Report

 In this study, the authors compared the outcomes of patients with advanced HCC whose tumors localized in liver who underwent hepatic arterial infusion chemotherapy (HAIC) with those who received sorafenib (SOR) in the cohort created by propensity score matching. Although this study provides interesting data that can be used as a reference for future studies to show the significance of HAIC for advanced HCC at a time when molecular targeted therapy and combined immunotherapy are becoming the mainstream of treatment for these diseases, there are some points that need to be improved.

Major points

  1. Why were patients with BCLC-A HCC included in this study, even though this study is to evaluate the efficacy of HAIC for highly advanced hepatocellular carcinoma? Shouldn't these patients be excluded from the study?
  2. Furthermore, patients with BCLC-B, the majority of the patients in this study, are conventionally treated with TACE as first-line therapy. The authors should describe the details of these patients, especially their tumor burden including up-to-7 in/out, tumor number, etc.
  3. Was there any crossover between the HAIC and SOR arms? In addition, the authors should describe the details of treatments used for these patients after 2nd line, including other molecular targeted therapies or combined immunotherapy. These are important factors that could affect the overall survival of both groups.
  4. The authors should describe about the details of adverse events occurred in patients.
  5. What did the authors want to show from the multivariate analysis? Is multivariate analysis necessary for this study?
  6. Changes in liver function during and after treatment in the HAIC and SOR groups should be shown, because maintenance of liver function is a critical factor in the treatment of advanced HCC and in prolonging the survival of these patients. Compared to patients in the SOR group, patients in the HAIC group may have maintained liver function better during and after treatment.
  7. “Eligibility criteria for this study were similar to those for the SHARP study” (P3L109) ->The details of eligibility criteria for this study should be described precisely.

Minor points

  1. The authors should describe methods of multivariate analysis in materials and methods section if they use.
  2. Similarly, the details of the propensity score matching method described in P5L161-169 should be described in the materials and methods section.

Author Response

In this study, the authors compared the outcomes of patients with advanced HCC whose tumors localized in liver who underwent hepatic arterial infusion chemotherapy (HAIC) with those who received sorafenib (SOR) in the cohort created by propensity score matching. Although this study provides interesting data that can be used as a reference for future studies to show the significance of HAIC for advanced HCC at a time when molecular targeted therapy and combined immunotherapy are becoming the mainstream of treatment for these diseases, there are some points that need to be improved.

Major points

  1. Why were patients with BCLC-A HCC included in this study, even though this study is to evaluate the efficacy of HAIC for highly advanced hepatocellular carcinoma? Shouldn't these patients be excluded from the study?

Response: We appreciate this thoughtful recommendation. We re-analyzed all patients, excluding those with BCLC-A HCC.

  1. Furthermore, patients with BCLC-B, the majority of the patients in this study, are conventionally treated with TACE as first-line therapy. The authors should describe the details of these patients, especially their tumor burden including up-to-7 in/out, tumor number, etc.

Response: We thank the reviewer for the helpful comment. Regarding the tumor number, 5 (6%) had single tumors and 83 (94%) had multiple tumors in the HAIC cohort, while 15 (6%) had single tumors and 228 (94%) had multiple tumors in the sorafenib cohort. Regarding the tumor size, it was 37.2 ± 30.9 mm in the HAIC cohort and 42.4 ± 23.2 mm in the sorafenib cohort.

  1. Was there any crossover between the HAIC and SOR arms? In addition, the authors should describe the details of treatments used for these patients after 2nd line, including other molecular targeted therapies or combined immunotherapy. These are important factors that could affect the overall survival of both groups.

Response: We thank the reviewer for this suggestion. There was no crossover between the HAIC with cisplatin and sorafenib cohorts. There was no secondary treatment in the HAIC with cisplatin cohort. Meanwhile, in the sorafenib cohort, 11 patients received other molecular-targeted agents and 32 patients received HAIC or TACE as the secondary treatment.

  1. The authors should describe about the details of adverse events occurred in patients.

Response: We thank the reviewer for this comment. There were few noticeable adverse events leading to discontinuation of treatment in the HAIC with cisplatin cohort—one case had anaphylactic shock as an adverse event that led to discontinuation of treatment. On the other hand, in the sorafenib cohort, 73 patients (88%) had any adverse events, comprising 28 patients with hand-foot skin reactions, 17 with liver dysfunction, 15 with diarrhea, 10 with loss of appetite, and 5 with hypertension (overlapped).

  1. What did the authors want to show from the multivariate analysis? Is multivariate analysis necessary for this study?

Response: We appreciate this recommendation and have revised Table 3.

  1. Changes in liver function during and after treatment in the HAIC and SOR groups should be shown, because maintenance of liver function is a critical factor in the treatment of advanced HCC and in prolonging the survival of these patients. Compared to patients in the SOR group, patients in the HAIC group may have maintained liver function better during and after treatment.

Response: We thank the reviewer for this suggestion. We compared the ALBI scores before treatment and 1 month after treatment as a simple index of hepatic reserve factor. In the HAIC with cisplatin cohort, it was -2.09 ± 0.48 before treatment and -2.08 ± 0.57 at 1 month after treatment (p = 0.9429). Meanwhile, in the sorafenib cohort, it was -2.14 ± 0.44 before treatment and -2.03 ± 0.52 at 1 month after treatment (p = 0.1792). There was no significant difference in the changes in hepatic reserve factor before and after treatment between the two cohorts.

  1. “Eligibility criteria for this study were similar to those for the SHARP study” (P3L109) ->The details of eligibility criteria for this study should be described precisely.

Response: We described the details of eligibility criteria in the Materials and Methods section as follows: 1) Eastern Cooperative Oncology Group performance status of 0–1, 2) measurable disease using the Response Evaluation Criteria in Solid Tumors, 3) Child-Pugh class A or B, 4) leukocyte count ≥ 2,000/mm3, 5) platelet count ≥ 50 × 109/L, 6) hemoglobin level ≥ 8.5 g/dL, 7) serum creatinine level < 1.5 mg/dL, and 8) no ascites or encephalopathy.

Minor points

  1. The authors should describe methods of multivariate analysis in materials and methods section if they use.

Response: We described the methods of the univariate and multivariate analyses in the Materials and Methods section of the revised manuscript.

  1. Similarly, the details of the propensity score matching method described in P5L161-169 should be described in the materials and methods section.

Response: We also described the details of propensity score matching in the Materials and Methods section of the revised manuscript.

Reviewer 3 Report

The authors compared prognostic factors for OS following HAIC with cisplatin (n=97) versus sorafenib (n=251) for intrahepatic advanced HCC using propensity score-matched analysis. Their findings suggest that HAIC with cisplatin demonstrates longer prognostic effects than sorafenib in intrahepatic advanced HCC, regardless of the hepatic reserve.

The manuscript is interesting. But I have some major concerns.

Major comments

#1. The information of previous therapy before HAIC or sorafenib should be described, since they are probable to affect the prognosis of the patients.

#2. The total dose and dose reduction of cisplatin or sorafenib, and the numbers of HAIC should be described, since they are probable to affect the prognosis of the patients.

#3. They should mention the therapy undertaken after tumor progression occurred during treatment of HAIC or sorafenib.

#4. They should present univariate and multivariate Analyses of OS in the sorafenib Cohort.

Author Response

The authors compared prognostic factors for OS following HAIC with cisplatin (n=97) versus sorafenib (n=251) for intrahepatic advanced HCC using propensity score-matched analysis. Their findings suggest that HAIC with cisplatin demonstrates longer prognostic effects than sorafenib in intrahepatic advanced HCC, regardless of the hepatic reserve.

The manuscript is interesting. But I have some major concerns.

Major comments

#1. The information of previous therapy before HAIC or sorafenib should be described, since they are probable to affect the prognosis of the patients.

Response: We appreciate this thoughtful recommendation. Thirteen patients received PRFA, 2 received PEIT, 16 received HAIC with the reservoir system, 25 received TACE, and 7 received hepatic resection as the previous treatment in the sorafenib cohort. Information on previous therapy before HAIC could not be obtained.

#2. The total dose and dose reduction of cisplatin or sorafenib, and the numbers of HAIC should be described, since they are probable to affect the prognosis of the patients.

Response: We thank the reviewer for this comment. The total dose (mg) in the HAIC cohort was 308.9 ± 318.3 and that in the sorafenib cohort was 69929.1 ± 97478.5. The number of HAIC cycles was 5.2 ± 4.8.

#3. They should mention the therapy undertaken after tumor progression occurred during treatment of HAIC or sorafenib.

Response: We thank the reviewer for this suggestion. There was no cases of secondary treatment in the HAIC cohort; on the other hand, there were 11 patients who received other molecular-targeted agents and 32 patients who received HAIC with the reservoir system or TACE, as the secondary treatment in the sorafenib cohort.

#4. They should present univariate and multivariate Analyses of OS in the sorafenib Cohort.

Response: We thank the reviewer for careful consideration of our data presentation. Univariate and multivariate analyses of OS following propensity score-matched analysis were re-analyzed in Table 3.

Round 2

Reviewer 2 Report

The authors have responded sincerely to the reviewers' suggestions and the analysis has been done appropriately. This manuscript has been now suitable for publication.

Author Response

We are very grateful for your comments.

Reviewer 3 Report

The authors revised the manuscript according to my comments satisfactorily. But Table 3 is not appropriately arranged, and should be revised.

Author Response

We are very grateful for your comments.

We have presented univariate and multivariate analyses of OS in the sorafenib cohort.